# Synthesis and Characterization of New Boron Compounds Using Reaction of Diazonium Tetraphenylborate with Enaminoamides

**DOI:** 10.3390/molecules27020367

**Published:** 2022-01-07

**Authors:** Markéta Svobodová, Jan Svoboda, Bing-Han Li, Valerio Bertolasi, Luboš Socha, Miloš Sedlák, Lukáš Marek

**Affiliations:** 1Faculty of Chemical Technology, Institute of Organic Chemistry and Technology, University of Pardubice, Studentská 573, 532 10 Pardubice, Czech Republic; jan.svoboda@upce.cz (J.S.); st46953@upce.cz (L.S.); milos.sedlak@upce.cz (M.S.); lukas.marek3@student.upce.cz (L.M.); 2Department of Chemistry, National Tsing Hua University, Hsinchu 30013, Taiwan; w6520325@hotmail.com; 3Department of Chemical and Pharmaceutical Sciences, University of Ferrara, Via L. Borsari 46, 44121 Ferrara, Italy; m38@unife.it

**Keywords:** boron, enaminoamide, oxazaborine, diazaborinone, triazaborine, triazaborinone

## Abstract

A family of oxazaborines, diazaborinones, triazaborines, and triazaborinones was prepared by reaction of polarized ethylenes, such as β-enaminoamides, with 4-methylbenzenediazonium tetraphenylborates. The reaction conditions (stirring in CH_2_Cl_2_ at room temperature (Method A) or stirring with CH_3_COONa in CH_2_Cl_2_ at room temperature (Method B) or refluxing in the CH_2_Cl_2_/toluene mixture (Method C)) controlled the formation and relative content of these compounds in the reaction mixtures from one to three products. Substituted oxazaborines gradually rearranged into diazaborinones at 250 °C. The prepared compounds were characterized by ^1^H NMR, ^13^C NMR, IR, and UV–Vis spectroscopy, HRMS, or microanalysis. The structure of individual compounds was confirmed by ^11^B NMR, ^15^N NMR, 1D NOESY, and X-ray analysis. The mechanism of reaction of enaminoamides with 4-methylbenzenediazonium tetraphenylborate was proposed.

## 1. Introduction

Activated alkenes, also known as polarized ethylenes, are molecules containing a double bond with the electron-withdrawing group (–C=O, –CN, –NO_2_, etc.) at one end and electron-donating groups (typically amino groups) at the other. A molecule with such an arrangement of groups is referred to as “*push–pull*” ethylene. Due to their versatile reactivity, they are important building blocks for the syntheses of a wide variety of different heterocyclic systems with practical usability. β-Enaminoamides **1** (Figure 1) also belong among representatives of the polarized ethylenes. The oxygen of the amide function donates the free electron pair, and the amino group in β-position of the double bond is able to withdraw it.

Compounds **1** have been used for the synthesis of uracil derivatives that inhibit histone deacetylase (HDAC), and these compounds have been efficient in inhibiting human recombinant (hr) HDAC1 [1]. Compounds **1** have been also used in the synthesis of dihydropyridines [2,3,4,5,6], and these dihydropyridines have been tested in cloned human α adrenoceptors, as well as the rat L-type calcium channel [4] or platelet activating factor (PAF) antagonist activity was measured [3]. Enaminoamides **1** can be used in the synthesis of other types of heterocycles, for example, pyridines [7], pyridine-2-one [8], pyrimidines [9], pyrimidinones [10,11,12,13], spirocyclohexane [14], 1*H*-pyrrole-2,3-dione [15], 2-oxopyrrole [11], pyrazolone [16], and isothiazole [17]. The coordination ability of these simple compounds has not been studied yet. The first mention is from November 2017 [18,19], where the authors tested prepared oxazaborines (Figure 1) for inhibition of the NLRP3 inflammasome in vitro and in vivo without affecting Ca^2+^ homeostasis.

We have studied the reactions of polarized ethylenes, such as β-enaminones, β-enaminonitriles, and β-enaminoamides, with substitution on the amino group, with diazonium salts, leading to heterocyclic compounds. These compounds are pyridazines [20,21,22], pyrazoles [23,24,25,26], oxazaborines, diazaborinones, and triazaborines [27,28,29,30,31]. In 2009, we published work on the reaction of β-enaminoamides **1** (Figure 1, R^3^ = Ph) and 4-methylbenzenediazonium tetraphenylborate to give compounds **I**–**III** (Figure 2) [28]. The ring transformation of compound **I** to diazaborinones **II** (predominantly when R^1^ = Me) and triazaborines **III** (predominantly when R^1^ = H) was proceeded under reflux in *N*,*N*-dimethylformamide [28]. We decided to change the substitution at the nitrogen atom of the amide group of enaminoamides as the basis for this paper. In the new study on enaminoamide reactivity, we report on synthetic transformations leading to N–B–O, N–B–N, and newly also N–N–B–N–C(=O) heterocyclic systems using the reactions of β-enaminoamides **1** (Figure 1, R^3^ = H, Me) and diazonium salts.

## 2. Results and Discussion

### 2.1. Synthesis

We prepared eight enaminoamides **1a**–**h** (Appendix A). The synthesis proceeded via an addition–elimination mechanism of the corresponding oxoamides with ammonia or methylamine. Oxoamides (except acetoacetamide (**4a**)) were obtained in two (compound **4b** [32]) or three (compounds **4c** and **4d**) steps from ethyl benzoylacetate and ethyl acetoacetate, respectively. Compound **4b** was prepared according to G. A. M. Giardina [32]. For the preparation of oxoamides **4c** and **4d**, the synthetic procedure according to B. Štefane and S. Polanc was used [33]. This method consists of activation of the carboxy group with BF_3_–Et_2_O, substitution of the alkoxy group, and hydrolysis to give oxoamide **4c** and **4****d,** or aminolysis to give enaminoamide **1h**.

Prepared enaminoamides **1a**–**h** reacted with 4-methylbenezenediazonium tetraphenylborate under three different conditions: (1) Stirring in CH_2_Cl_2_ at room temperature (Method A), (2) Stirring with CH_3_COONa in CH_2_Cl_2_ at room temperature (Method B), and (3) Refluxing in the CH_2_Cl_2_/toluene mixture (Method C).

Figure 2 shows five possible product structures (A–E) for the reaction of enaminoamides **1a**–**h** with 4-methylbenzenediazonium tetraphenylborate.

When β-enaminoamide **1c** (Figure 1, R^1^ = R^2^ = Me, R^3^ = H) reacted with diazonium salt under heating with the mixture solvent (dichloromethane/toluene), the only product with a yield of 69% was isolated. The structure of this product was deduced on the basis of ^1^H, ^13^C, ^11^B, and ^15^N-NMR spectroscopy. In the aliphatic part of the ^1^H NMR spectrum of this product (Appendix A), there are three signals of three methyl groups. One of them is a doublet, and two of them are singlets. The presence of a doublet is only possible for structures **D** and **E** (Figure 2, R^1^ = R^2^ = Me, R^3^ = H). The 1D NOESY spectrum (Figure 3) helped us to exclude one of these suggested structures. The through-space interaction between the proton with δ = 5.52 ppm and the *ortho* protons of the phenyl groups on the boron atom was found after the selective excitation of this N–H proton (An analogue experiment was also carried out in the case of compounds **5a**, **5b**, **5g**. Appendix A). This interaction is not possible for structure **E**. The structure of triazaborinone **D** suggested on the basis of NMR experiments was subsequently confirmed by X-ray diffraction (Figure 4). This is the new type of heterocycle in the series of enaminoamides **1**, where the substituent on the nitrogen of the amide group is not phenyl. In the case of *N*-phenylenaminoamides, triazaborinones were not isolated.

We isolated the new type of boron heterocycle in six cases. When enaminoamides **1a**–**d**, **1g**, and **1h** reacted with diazonium salt, the new triazaborinones **5** were obtained as major compounds together with oxazaborines **6**, diazaborinones **7**, or triazaborines **8** as minor products (Figure 3). Oxazaborine **6a** was also prepared from 3-aminobut-2-ennitrile and diazonium tetraphenylborate, as described in [30].

When enaminoamides **1e** and **1f** reacted with diazonium tetraphenylborate, only oxazaborines and diazaborinones were isolated (Figure 4). Triazaborine **8e** was also isolated when enaminoamide **1e** and diazonium salt were refluxed after stirring 30 min at room temperature. The yield of **8e** was 26%.

The yields of the prepared heterocyclic boron compounds are shown in Table 1, and their values are given either after crystallization or after washing with an appropriate solvent.

We suggested the mechanism for the reaction of enaminones with diazonium tetraphenylborates [27]. In the first step, the azocoupling reaction proceeds in which tetraphenylborate acts as a base and during which the protodeboronation of Ph_3_B^–^–[C_6_H_6_]^+^ occurs to give triphenylborane and benzene [27]. Triphenylborane is able to coordinate to the nitrogen atom of the amine group of enaminones followed by further protodeboronation and cyclization. In the case of enaminoamides, two nitrogen atoms are possible for coordination (amine group and amide group) of triphenylborane. Our previously studied enaminoamides having phenyl at amidic nitrogen (Figure 1, R^3^ = Ph) gave products of coordination of triphenylborane with the amine group of enaminoamides (Figure 5, Path B). Enaminoamides **1** described in this paper gave different results. Triphenylborane likely bonded to amidic nitrogen (Figure 5, Path A), and after protodeboronation and cyclization, the new type of heterocyclic compounds formed. Triazaborinones **5** are the main products in most cases. The by-product was mainly diazaborinone. Exceptions are reactions of enaminoamides **1e** and **1f**, having –CONHMe, and the primary amine group. The behavior of these compounds is similar to that of enaminoamides having –CONHPh studied earlier. Oxazaborines **6e** and **6f**, diazaborinones **7e** and **7f**, and triazaborine **8e** were isolated when these enaminoamides were used.

Additionally, the products of double coordination of BPh_3_ (**9g** and **9h**, Figure 5) were isolated in the case of the reaction of enaminoamides **1g** and **1h** with diazonium tetraphenylborate carried out in dichloromethane at room temperature (Method A) for 72 h and 96 h, respectively. The yield was 4% for **9g** and 15% for **9h**. Compound **9h** was also obtained using the Method C (refluxing the mixture for 3 h), but the yield was only 3%. Chemical shifts of ^11^B and ^15^N are similar to those for oxazaborine **6h** and triazaborinone **5h**. The study of this type of structure will be the subject of further investigation.

When oxazaborines **6a** and **6e**–**h** were heated at 250 °C without solvent (Figure 6), a ring transformation occurred to give the corresponding diazaborinones in good yields (49–82%). Diazaborinone **7a** was not obtained from enaminoamide **1a** and diazonium salt, while recyclization gave compound **7a** in 82% yield. Here is the difference between the previously prepared oxazaborines **I** (Figure 2) with the *N*-phenyl group and those in this publication, since only diazaborinones were isolated, and it does not matter what the substitution of the amino group of the starting enaminoamide is.

NMR parameters such as ^15^N and ^11^B NMR chemical shifts (δ) and coupling constants (*J*) for heterocyclic compounds **5**–**9** are shown in Table 2.

### 2.2. UV–Visible Spectroscopy

UV–Vis analyses of the compounds **5g**, **6g**, and **7a** in various solvents (cyclohexane, diethyl ether, dichloromethane, tetrahydrofurane, acetonitrile, methanol, formamide) are shown in Figure 6 (**5g**, **7a**) and Appendix A (**6g**). All absorption spectra for prepared compounds in dichloromethane are listed in the Appendix A). The results of UV–Vis absorption data as an evaluated molar absorption coefficient (ε) for individual compounds in the solvents used are presented in Appendix A. The dependence of the maximum absorption spectra (λ_MAX_) on the solvent properties is not apparent from the above measurements. The possible solvatochromic effects are therefore rather insignificant.

### 2.3. IR Spectroscopy

The set of measured IR spectra for triazaborinones **5** is depicted in Figure 7, for oxazaborines **6** in Figure 8, for diazaborinones **7** in Figure 9, and for triazaborines **8** and oxazaborine–triazaborines **9** in Figure 10. Weak stretching vibrations of the C–H bonds of aromatic benzene rings at 3080–3000 cm^−1^ are evident in all spectra. Next, it is possible to mention the adjacent H–H out-of-plane deformation at the *para*-substituted ring at 820 cm^−1^ and several strong vibrations of the monosubstituted benzene ring, the out-of-plane ring bending at 706 cm^−1^, and the adjacent H–H out-of-plane deformation at 746 cm^−1^.

IR spectra **5a** and **5b** show the stretching vibration of the primary amino group at 3362 cm^−1^. The stretching vibration of the N–H and C=O groups in the *cis* configuration is evident for spectra **5a**–**d** at 3200 and 1640 cm^−1^, respectively. Strong stretching vibrations of the C=O of α,β-unsaturated β-amino-substituted imide are located in the 1630–1580 cm^−1^ region. In this case, the effect of substitution on the double bond is evident. The methyl-substituted (R^1^ = Me) compounds **5a**, **5c**, and **5g** are shifted to higher wavenumbers than the phenyl-substituted (R^1^ = Ph) compounds **5b**, **5d**, and **5h**. The sharp band caused by the vibration of the ring at 1430 cm^−1^ is characteristic of compounds containing a B–phenyl bond. The ring C–H out-of-plane deformation for phenyl-substituted boron at 760 cm^−1^ is present as a shoulder. Two phenyl groups should appear as a doublet with a 20 cm^−1^ separation. These bands can be overlayed by the adjacent H–H out-of-plane deformation at 746 cm^−1^, as mentioned above.

Figure 8 shows the band at 3350 cm^−1^, which corresponds to the free N–H stretching of the imine bound to the oxazaborine ring. This is consistent with the fact that this vibration occurs only for compounds **6a**, **6e**, and **6f**. In the case of **6a**, this band is broadened due to the stretching vibration of the primary amino group.

The bands of the N–H stretching vibrations of the imine bonded to the diazaborinone ring at 3350 cm^−1^ for **7a**, **7b**, **7e**, and **7f** are shown in Figure 9. The stretching vibration of the N–H (spectra **7a**–**d**) and C=O (spectra **7a**–**h**) groups in the *cis* configuration is evident in the range of 3200–3100 cm^−1^ and 1650–1620 cm^−1^, respectively. The spectra of all diazaborinones **7a**–**h** contain very strong bands in the 1530–1510 cm^−1^ region, corresponding to an arrangement of atoms in a hydrazoketones (R^1^–CO–CR^2^=N–NH–). For oxazaborines **6**, Figure 8 shows two strong absorption bands—one at approximately 1615 cm^−1^ and another more intense near 1580 cm^−1^—due to vibrations of the conjugated system of the unsaturated double bond with the aromatic diazo group.

Triazaborine **8e** (Figure 10) shows bands at 3440 and 3414 cm^−1^, which correspond to the free N–H stretching of imine bonded to the triazaborine ring. The stretching vibration of the N–H and C=O groups in the *cis* configuration is evident for spectra **8e** and **8h** in the range of 3290–3180 cm^−1^ and 1640–1626 cm^−1^, respectively. The position of these bands is affected by methyl- or phenyl-substitution adjacent to the amide. The sharp band caused by the ring vibration at 1430 cm^−1^ is characteristic of compounds containing a B–phenyl bond and is evident in all cases in the Figure 10. The spectra of triazaborines **8g**, **8e**, and **8h** contain very strong bands at 1510 cm^−1^, which corresponds to an arrangement of atoms in a hydrazoketones (R^1^–CO–CR^2^=N–NH–).

The other selected absorption bands in Figure 8, Figure 9 and Figure 10 are analogous to those in Figure 7 and are described above.

### 2.4. X-ray

For compounds **5c**, **5d**, **5g**, **6a**, **6e**, and **7f**, we were able to prepare crystals suitable for single-crystal X-ray diffraction. The results are summarized in Appendix A.

Triazaborinone **5c** contains two types of hydrogen bonds. The first intramolecular bond is located between the N1 and O1 atoms. The other two intermolecular hydrogen bonds between N4 from the first molecule and O1 from the second molecule and vice versa bind the dimer-forming molecules. See Figure 11a for details. The plane passing through the atoms forming these intermolecular hydrogen bonds is shown in this Figure. The center of gravity of all dimer atoms is in the center of the hydrogen bond system. The point is marked as a centroid. The periodic arrangement of these basic dimers in the crystal is shown in Appendix A. The view in the direction of the crystallographic axis *a* and *c*, and the reciprocal cell axis *b** is shown. The conjugate system of three double bonds (between O1 and C1, C2 and N2, and N1 and C3 in one molecule) lies in a plane. The analogous plane from the second molecule of the dimer is parallel to the first plane and is shifted by 0.128 Å, as can be seen in Appendix A. Thus, all of these double bonds in the dimer lie practically in the plane. These bonds were assigned as doubles on the basis of their lengths, and the fact that they lie in a plane supports the indicated structure with a lower application of conjugation between the triazaborinone ring and the aromatic ring. The aromatic benzene rings are deflected out of these planes as well as boron atoms.

Triazaborinone **5d** forms dimers such as **5c**. The structure of the dimer is shown in Figure 11b. As in the previous case, the dimer molecule is symmetric according to the center of symmetry, which in this case is at the beginning of the coordinates of the unit cell. By replacing the methyl group bound to carbon C3 with a phenyl group, hydrophobic cavities are formed in the structure of triazaborinone **5d** in which toluene is enclosed as a solvent, one molecule of toluene stoichiometrically per dimer **5d**. The toluene molecule is captured in the middle of the *bc* side of the crystallographic unit cell and is green-colored for clarity. The toluene molecule is located freely in the cavity. The presence of a toluene methyl group on two sides is a disorder for statistical reasons. Similar host–guest structures have been previously described by us for 3-amino-2-(4-dimethylaminophenyldiazenyl)-1-phenylbut-2-en-1-one (reaction of 3-amino-1-phenylbut-2-en-1-one with 4-dimethylaminobenzenediazonium tetrafluoroborate) [34]. The arrangement of the molecules in the crystal is shown in Appendix A. The hydrogen bonds that form the dimers are again shown in the figures for easier orientation.

Triazaborinone **5g** has a similar arrangement of double bonds lying in the plane as triazaborinone **5c**. The ORTEP diagram is shown in Figure 12a. Because of the substitution of imide hydrogen on the triazaborinone ring with a methyl group, triazaborinone **5g** cannot form intramolecular hydrogen bonds leading to dimers. In this case, the crystal structure is formed by individual molecules that are bound together by weak nonbonding interactions, e.g., between aromatic rings. This arrangement is shown in Appendix A. The interaction mentioned can be seen in the middle of an indicated crystallographic unit cell. These phenyl groups attached to the boron atom form graphite-like π–π stacking with a distance between these rings of 3.293 Å. The chains formed by this layering in the direction of the crystallographic axis *b* are repeated regularly in the directions of the crystallographic axes *a* and *c*. The planes indicated in the figure are those in which the conjugated double bonds between the atoms C1=O1, C2=N2, and C3=N4 lie. All atoms of the triazaborinone ring of compound **5g** lie in the plane from which the ring of the *p*-tolyl group is slightly turned out.

Oxazaborine **6a**, which was previously prepared from 3-aminobut-2-ennitrile as the starting compound, was characterized in our previous publication [30]. Unlike our earlier publication, the prepared single crystal **6a** was now measured at room temperature. The structure obtained is identical, and the newly estimated values are given in Appendix A. The new parameters slightly differ from those previously published due to the measurement temperature, and the resulting statistics are slightly better. Unlike other prepared single crystals at this work, only oxazaborine **6a** needs two molecules to form an asymmetric unit (see Figure 12b or Appendix A). In addition, in the case of this substance, three conjugated double bonds lie in the plane. They are N3=N4, C1=C2, and C3=N2. In this case, the boron atoms are completely deflected out of these planes, and the N–B–O angles are 103.69° and 104.13°. Sizes of such angles are not common but are described in the literature.

Oxazaborine **6e** differs from previously published **6a** by substituting the hydrogen atom of the amide with a methyl group. Although the structure of oxazaborine **6a** consists of dimers due to hydrogen bonds similar to the structure of triazaborinone **5c** described above, the formation of such dimers in the structure of oxazaborine **6e** is not possible. The oxygen atom O1 in compound **6e** does not participate in the formation of any hydrogen bond. In this case, the entire oxazaborine ring lies in the plane, as well as the *p*-tolyl group bonded to nitrogen N4. Benzene rings attached to a boron atom outside this plane form a roughly tetragonal system. The ORTEP diagram of this compound shows Figure 13a. The crystal structure of oxazaborine **6e** is formed by chains of individual molecules bound to each other by π–π interactions. These interactions in the direction of the crystallographic axis *c* are shown in Appendix A. Appendix A shows the alternation of these chains in the direction of the crystallographic axes *a* and *c*.

The ORTEP diagram of the diazaborinone **7f** is shown in Figure 13b. Two similar diazaborinones have been previously published [28]. The first differs by the substitutions on the C1 and N2 atoms, where the substituents are exchanged. The second differs by the substitution on the N2 atom, where the phenyl group is replaced by a methyl group. In the structures described previously, the formation of chains of individual molecules was observed due to the short distance interaction between the oxygen atom O1 and the imine hydrogen bonded to the nitrogen atom N1 of the second molecule. In the case of the newly described structure of diazaborinone **7f**, such an intermolecular hydrogen bond does not occur, and only weak interactions occur between the molecules in a crystal. The crystal packing in the direction of the crystallographic axes *a* and *b* is shown in Appendix A. The first interaction is a π–π stacking between the *p*-tolyl groups. These interactions lie on the *ab* side of the crystallographic unit cell. The second interaction is between phenyl groups bonded to boron atoms. These groups form layers that are nested inside each other. These interactions lie in the center plane of the crystallographic unit cell in one-half of the edge length *c*.

### 2.5. Fluorescence

Some substances showed visible fluorescence when irradiated with a UV lamp (366 nm). This was found during thin-layer chromatography with UV detection. Therefore, steady-state photoluminescence measurements in the solid state were performed for a series of triazaborinones **5a**–**d**, **5g**, and **5h**; diazaborinones **7a** and **7f**–**g**; triazaborine **8h**; and several previously prepared substances. The measured substances were excited by radiation at a wavelength of 360 nm, and the corresponding spectra are shown in Figure 14 and Figure 15.

Time-resolved photoluminescence measurements were also performed, and quantum yields were evaluated. Unfortunately, measurable quantum yields were observed for only three substances. Fluorescence quantum yields values are not very high. The quantum yield for triazaborinone **5h** is 3.84% ± 1%, for triazaborinone **5g** 4.08% ± 1%, and for diazaborinone **7a** 14.62% ± 1%. The absorption and emission spectra for these substances are shown in Appendix A.

## 3. Experimental

### 3.1. Materials and Methods

All chemicals and dried dichloromethane except those mentioned below were purchased from commercial suppliers (Acros Organics (Part of Thermo Fisher Scientific, Geel, Belgium)), Sigma-Aldrich (Merck, Darmstadt, Germany), or Fluorochem (Hadfield, Derbyshire, UK). Toluene and diethylether were dried over sodium.

NMR spectra were measured in CDCl_3_ or in DMSO-*d*_6_ using a Bruker AVANCE III 400 spectrometer (Ettlingen, Germany) operating at 400 MHz (^1^H) and 100 MHz (^13^C) or using a Bruker Ascend 500 spectrometer (Ettlingen, Germany) operating at 500 MHz (^1^H), 125 MHz (^13^C), 160 MHz (^11^B), and 50 MHz (^15^N). All of the pulse sequences were taken from the Bruker software library. The ^13^C NMR spectra were measured in a standard way and by means of the APT pulse sequence. The data are reported as follows: chemical shift in ppm (δ), multiplicity (s = singlet, d = doublet, q = quartet, m = multiplet, br s = broad singlet, br q = broad quartet). The coupling constants *J* are reported in Hertz (Hz). TMS was used as an internal standard for ^1^H NMR in CDCl_3_ (δ 0.00). CDCl_3_ was used as an internal standard for ^13^C NMR (the middle signal, δ 77.16). B(OMe)_3_ was used as an external standard for ^11^B NMR (δ 18.1). Nitromethane was used as an external standard for ^15^N NMR (δ 0.00). DMSO-*d*_6_ was used as an internal standard for both ^1^H NMR (the middle signal δ 2.55) and ^13^C NMR (the middle signal δ 39.6).

Elemental analyses were performed on a Flash 2000 CHNS Elemental Analyzer (Thermo Fisher Scientific, Milan, Italy).

Melting points were measured on a Kofler Boetius PHMK 80/2644 hot-stage microscope and were not corrected.

High-resolution mass spectra were recorded on a MALDI LTQ Orbitrap XL (Thermo Fisher Scientific, Bremen, Italy) equipped with a nitrogen UV laser (337 nm, 60 Hz, 8–20 μJ) in the positive ion mode. For the CID experiment using the linear trap quadrupole (LTQ) helium was used as the collision gas and 2,5-dihydroxybenzoic acid (DHB) as the MALDI matrix.

UV−Vis spectra were recorded on a UV–Vis spectrophotometer Hewlett-Packard 8453 (Waldbronn, Germany). IR spectra were recorded on a Nicolet iS50 (Madison, Wisconsin, USA) equipped with an ATR diamond crystal (neat solid samples). The wavenumber range 2500–1700 cm^−1^ was excluded due to diamond absorption. The spectra were processed with the SPECTRAGRYPH 1.2.15 (Dr. Friedrich Menges, Oberstdorf, Germany).

Steady-state and time-resolved PL measurements were performed based on a multifunctional spectrometer with L-geometry of excitation-detection pathways (FluoTime 300, PicoQuant, Berlin, Germany). A Xenon arc lamp and a PMT were used as the excitation sources and detectors for PL experiments. The PL-QY value could be obtained based on the spectrometer incorporated with an integrating sphere. Excitation wavelength: 360 nm. Range: 340–700 nm.

Crystal data for all compounds were collected at 295 K using a Nonius Kappa CCD diffractometer with graphite monochromated Mo-Kα radiation (λ = 0.71073 Ǻ). The data sets were integrated with the Denzo SMN package [35] and corrected for Lorentz, polarization, and absorption effects (SORTAV) [36]. Structures were solved by direct methods using the SIR97 [37] system of programs and refined anisotropically by using full-matrix least-squares for all non-hydrogen atoms and hydrogen atoms included on their calculated positions, riding on their carrier atoms, except the N–H hydrogens forming intramolecular hydrogen bonds, which were refined isotropically. All calculations were performed by using SHELXL 2014/6 [38], PARST [39] and PLATON [40] implemented in the WINGX [41] system of programs. Visualization of structures was done with MERCURY 2020.2.0.

### 3.2. Synthetic Procedure

*3-Aminobut-2-enamide* (**1a**). The dry ammonia gas was bubbled into a mixture of 3-oxobutanamide (**4a**) (10.1 g, 100 mmol) and diethyl ether (20 mL) over a period of 1 h. Then liquid ammonia (15 mL) was added and the mixture was stirred at room temperature overnight. The crystals were filtered off and boiled in chloroform (30 mL) for 5 min. After cooling, the white crystals were filtered off. Yield 7.13 g (71%), m.p. 95.5–99 °C (Ref. [42] 98–100 °C). ^1^H-NMR (DMSO-*d*_6_, 500 MHz): δ 7.31 (br s, 2H), 6.45 (br s, 1H), 6.09 (br s, 1H), 4.35 (s, 1H), 1.76 (s, 3H); ^13^C-NMR (DMSO-*d*_6_, 125 MHz): δ 172.6, 157.0, 85.4, 25.7, 21.8. 

*3-Amino-3-phenylprop-2-enamide* (**1b**). The solution of ethyl 3-oxo-3-phenylpropanoate (22.1 g, 115 mmol) and 25% aq. ammonia (22.7 mL, 300 mmol) was heated with stirring in an Ace pressure tube at 130 °C (oil bath) for 4.5 h. After cooling, the solid product was filtered off. Yield 6.6 g. The filtrate was concentrated and then aq. ammonia (16 mL) was added and heated in an Ace pressure tube at 130 °C for 3 h. After cooling, the solid product was filtered off. Yield 1.5 g. Yellowish crystal, total yield 8.1 g (43%), m.p. 162–164.5 °C (Ref. [16] 164 °C). ^1^H-NMR (DMSO-*d*_6_, 400 MHz): δ 7.58–7.60 (m, 2H), 7.46–7.48 (m, 3H), 6.85 (br s, 1H), 6.26 (br s, 1H), 4.91 (s, 1H); ^13^C-NMR (DMSO-*d*_6_, 100 MHz,): δ 172.2, 157.1, 137.9, 129.6, 128.6, 126.1, 85.9.

*3-(Methylamino)but-2-enamide* (**1c**). 3-Oxobutanamide (**4a**) (5 g, 50 mmol) and methylamine solution (33 wt.% in absolute ethanol, 18.5 mL, 150 mmol) were heated in an Ace pressure tube for 4 h at 130–140 °C. After cooling to room temperature, ethanol was evaporated, and the oil was cooled. Ethyl acetate (5 mL) was added, and the mixture was stirred for 5 min at cooling. The crystals were filtered off, and the filtrate was evaporated. Ethyl acetate (3 mL) was added to the filtrate and the mixture was stirred for 5 min at cooling. The crystals were filtered off. This compound must be stored in the refrigerator. Total yield 3.33 g (58%), m.p. 70–74 °C. ^1^H-NMR (CDCl_3_, 400 MHz): δ 8.99 (br s, 1H), 5.01 (br s, 2H), 4.37 (s, 1H), 2.87 (d, *J* = 5.3 Hz, 3H), 1.87 (s, 3H); ^13^C-NMR (CDCl_3_, 100 MHz): δ 173.0, 161.0, 83.1, 29.3, 19.1. HR-MS (MALDI), cald. for C_5_H_11_N_2_O: [M + H]^+^ 115.0866, found: 115.0867 [M + H]^+^.

*3-(Methylamino)-3-phenylprop-2-enamide* (**1d**). The solution of 3-oxo-3-phenylprop-2-enamide (**4b**) (3.2 g, 19.6 mmol) and 33 wt% methylamine in ethanol (7.4 mL, 58.8 mmol) was heated with stirring in an Ace pressure tube at 135–140 °C. After heating for 4 h, ethanol was evaporated under reduced pressure. Ethyl acetate (5 mL) was added to the viscous oil and the mixture was heated for 2 min. After cooling, the light-yellow crystals were filtered off. Yield 2.8 g (81%), m.p. 93.5–96 °C. ^1^H-NMR (CDCl_3_, 500 MHz): δ 9.01 (br s, 1H), 7.38–7.39 (m, 3H), 7.32–7.34 (m, 2H), 5.09 (br s, 2H), 4.48 (s, 1H), 2.71 (d, *J* = 5.2 Hz, 3H); ^13^C-NMR (CDCl_3_, 125 MHz): δ 172.7, 164.2, 136.3, 129.0, 128.3, 127.9, 86.1, 31.3. For elemental analysis, the product was recrystallized from the cyclohexane/toluene mixture. Anal. Calc. for C_10_H_12_N_2_O: C, 68.16; H, 6.86; N, 15.90. Found: C, 68.40; H, 6.90; N, 15.87%.

*3-Amino-N-methylbut-2-enamide* (**1e**). The title compound was prepared according to a modified published procedure [43]. Dry ammonia gas was bubbled into a mixture of *N*-methyl-3-oxobutanamide (**4c**) (2.26 g, 19.6 mmol), dry toluene (10 mL), and molecular sieve (1 g) over a period of 2 h at 10 °C. Then ethanol (30 mL) was added. The solid was dissolved, the mixture was filtered off, and the filtrate was evaporated to dryness. Cold diethyl ether was added, and the mixture was stirred. Then the white crystals were filtered off. This compound must be stored in the refrigerator. Yield 1.59 g (71%), m.p. 95.5–99 °C (Ref. [43] 115–116 °C). ^1^H-NMR (CDCl_3_, 400 MHz): δ 6.33 (br s, 2H), 5.26 (br s, 1H), 4.36 (s, 1H), 2.78 (d, *J* = 4.9 Hz, 3H), 1.84 (s, 3H); ^13^C NMR (CDCl_3_, 100 MHz): δ 171.3, 156.0, 86.6, 25.6, 22.3. Anal. Calc. for C_5_H_10_N_2_O: C, 52.61; H, 8.81; N, 24.54. Found: C, 52.58; H, 8.93; N, 23.81%.

*3-Amino-N-methyl-3-phenylprop-2-enamide* (**1f**). The title compound was prepared according to the published procedure [44]. *N*-Methyl-3-oxo-3-phenylpropanamide (**4d**) (7 g, 40 mmol), ammonium acetate (16.3 g, 200 mmol) and methanol (100 mL) were stirred at 55 °C for 3 days. Then, aqueous ammonia (50 mL) was added. The reaction mixture was extracted with dichloromethane (2 × 50 mL). The organic layer was dried with anhydrous sodium sulphate, and solvent was removed by distillation. Dichloromethane (50 mL) was added to the residue, and the organic layer was again extracted with water (1 × 50 mL). The organic layer was dried with anhydrous sodium sulphate, and solvent was removed by distillation. Recrystallization from ethyl acetate was obtained 3.78 g (55%). White crystals, m.p. 127–130 °C. ^1^H-NMR (CDCl_3_, 400 MHz): δ 7.28–7.40 (m, 5H), 6.56 (s, 2H), 5.47 (s, 1H), 4.8 (s, 1H), 2.83 (d, *J* = 5.2 Hz, 3H); ^13^C-NMR (CDCl_3_, 100 MHz): δ 171.1, 157.3, 138.5, 129.7, 128.7, 126.1, 87.5, 25.9. Anal. Calc. for C_10_H_12_N_2_O: C, 68.16; H, 6.86; N, 15.90. Found: C, 68.24; H, 6.98; N, 15.71%.

*N-Methyl-3-(methylamino)but-2-enamide* (**1g**). *N*-Methyl-3-oxobutanamide (**4c**) (2.65 g, 23 mmol): 33 wt% solution of methylamine in ethanol (8.6 mL, 69 mmol) and montmorillonite K-10 (5.8 g) was boiled ca 60 min. After cooling, dichloromethane (10 mL) was added, and montmorillonite was filtered and washed with dichloromethane (4 × 20 mL). Dichloromethane was distilled off, and the residue solidified in the refrigerator overnight. Then the beige crystals were filtered off. This compound must be stored in the refrigerator. Yield 2.63 g (89%), m.p. 50–54 °C. ^1^H-NMR (CDCl_3_, 400 MHz): δ 5.87 (br s, 1H), 5.23 (br s, 1H), 4.33 (s, 1H), 2.85 (d, *J* = 5.2 Hz, 3H), 2.75 (d, *J* = 4.9 Hz, 3H), 1.86 (s, 3H); ^13^C-NMR (CDCl_3_, 100 MHz): δ 171.7 (br), 158.9 (br), 84.3 (br), 29.2, 25.5 (br), 19.0. HR-MS (MALDI), cald. for C_6_H_13_N_2_O: [M + H]^+^ 129.1022, found: 129.1023 [M + H]^+^.

*N-Methyl-3-(methylamino)-3-phenylprop-2-enamide* (**1h**). The method according to Štefane and Polanc was used [33]. The solution of **3b** (3.38 g, 15 mmol) and 33 wt% methylamine in ethanol (9.3 mL) in propan-1-ol (65 mL) was heated with stirring in an Ace pressure tube at 130–140 °C. After heating for 4 h, the reaction mixture was evaporated under reduced pressure, and the residue was cooled. Slightly yellowish crystals were filtered off. Yield 4 g (96%), m.p. 101–105 °C. ^1^H-NMR (CDCl_3_, 400 MHz): δ 8.86 (br s, 1H), 7.34–7.36 (m, 3H), 7.29–7.32 (m, 2H), 5.39 (br s, 1H), 4.45 (s, 1H), 2.79 (d, *J* = 4.9 Hz, 3H), 2.64 (d, *J* = 5.3 Hz, 3H); ^13^C-NMR (CDCl_3_, 100 MHz): δ 171.3, 162.4, 136.6, 128.7, 128.2, 127.9, 87.9, 31.2, 25.7. For elemental analysis, this compound was recrystallized from the cyclohexane/toluene mixture. Anal. Calc. for C_11_H_14_N_2_O: C, 69.45; H, 7.42; N, 14.73. Found: C, 69.58; H, 7.50; N, 14.70%.

*General procedure for the preparation of 4-ethoxy-2,2-difluoro-1,3,2-dioxoborinanes* **2a**, **b**. A modified method according to Štefane and Polanc was used [33]. To a solution of ethyl benzoylacetate (156 mmol, 30 g) in dichloromethane (35 mL) 48% BF_3_–Et_2_O (312 mmol, 39.6 mL) was added at 23–25 °C. After stirring at 23–25 °C for 24 h, the precipitated material was filtered off. All volatile materials were removed under vacuum from the filtrate. Cooled diethyl ether (100 mL) was added to the residue, and after stirring for 5 min, the crystals were filtered off. Filtrate was evaporated again, and the residue was stirred with a small amount of cooled diethyl ether. The precipitated crystals were filtered off. The total yield of white crystals was 26.95 g (72%). M.p. 105–106 °C (Ref. [45] 108–110 °C). For compound **2a**: After stirring at 23–25 °C for 26.5 h, all volatile materials were removed under vacuum, and the yellow oily product was used for the preparation of compound **3a** without purification.

*2,2-Difluoro-4-methyl-6-(methylamino)-1,3,2-dioxoborinane* (**3a**). The method according to Štefane and Polanc was used [33]. To a stirred solution of 33 wt% methylamine in ethanol (37 mL, 297 mmol) and acetonitrile (245 mL), crude dioxoborinane **2a** was added at 23 °C. The reaction mixture was stirred 30 min at 23 °C. The solvents were evaporated, and the residue was separated by column chromatography using dichloromethane as an eluent to give 16.85 g (45%). Light-yellow crystals, m.p. 95–99 °C. ^1^H-NMR (CDCl_3_, 400 MHz): δ 7.30 (br s, 1H, maj.), 6.51 (br s, 0,09H, min.), 5.36 (s, 1H, min.), 5.33 (s, 1H, maj.), 3.00 (d, *J* = 4,8 Hz, 3.26 H, maj. + min.), 2.16 (s, 0,30H, min.), 2.05 (s, 3H, maj.); ^13^C-NMR (CDCl_3_, 100 MHz): δ 181.1 (min.), 176.8 (maj.), 169.2 (min.), 168.8 (maj.), 87.8 (maj.), 84.0 (min.), 28.7 (min.), 27.2 (maj.), 23.3 (min.), 22.5 (maj.); ^19^F NMR (CDCl_3_, 376 MHz): δ −142.20 (^19^F–^11^B, 0.8F, min.), −142.14 (^19^F–^10^B, 0.2F, min.), –142.09 (^19^F–^11^B, 0.8F, maj.), –142.03 (^19^F–^10^B, 0.2F, maj.). For elemental analysis, this compound was recrystallized from ethyl acetate. Anal. Calc. for C_5_H_8_BF_2_NO_2_: C, 36.86; H, 4.95; N, 8.60. Found: C 37.12; H 5.07; N 8.63%.

*2,2-Difluoro-6-(methylamino)-4-phenyl-1,3,2-dioxoborinane* (**3b**). The method according to Štefane and Polanc was used [33]. To a stirred solution of 33 wt% methylamine in ethanol (18.1 mL, 146 mmol) and acetonitrile (118 mL), dioxoborinane **2b** (26.95 g, 112 mmol) was added at room temperature. The reaction mixture was stirred for 30 min at room temperature. The solvents were evaporated, and the residue was stirred in diethyl ether (40 mL). The crystals were filtered off. White crystals, yield 25 g (84%), m.p. 164–168 °C. ^1^H-NMR (DMSO-*d*_6_, 500 MHz): δ 8.07–8.08 (m, 0.21H, min.), 7.87–7.88 (m, 2H, maj.), 7.54–7.62 (m, 3.37H, maj. + min.), 6,42 (s, 0.1H, min.), 6.14 (s, 1H, maj.), 3.05 (s, 0.36H, min.), 2.96 (s, 3H, maj.); ^13^C-NMR (DMSO-*d*_6_, 125 MHz): δ 171.9 (min.), 169.0 (maj.), 168.9 (min.), 168.2 (maj.), 133.0 (min.), 132.9 (maj.), 132.6 (min.), 132.2 (maj.), 129.0 (maj.), 128.9 (min.), 126.6 (min.), 126.5 (maj.), 85.6 (maj.), 81.2 (min.), 28.5 (min.), 26.9 (maj.); ^19^F-NMR (DMSO-*d*_6_, 470 MHz): δ −141.77 (^19^F–^11^B, 0.8F, min.), −141.71 (^19^F–^10^B, 0.2F, min.), −141.49 (^19^F–^11^B, 0.8F, maj.), −141.44 (^19^F–^10^B, 0.2F, maj.). For elemental analysis, this compound was recrystallized from ethanol. Anal. Calc. for C_10_H_10_BF_2_NO_2_: C, 53.38; H, 4.48; N, 6.23. Found: C 53.68; H 4.60; N 6.23%.

*3-Oxo-3-phenylprop-2-enamide* (**4b**). This compound was prepared according to the procedure in Ref. [32]. 3-Amino-3-phenylprop-2-enamide (**1b**) (8 g, 49 mmol) was heated in water (33 mL) for 4 h. After cooling, the solid product was filtered off. Crystals were crystallized from water. Yellowish crystals, yield 4.62 g (57%), m.p. 108–111 °C (Ref. [32] 110–111 °C).

*N-Methyl-3-oxobutanamide* (**4c**). This compound was prepared according to the procedure in Ref. [33]. The mixture of 2,2-difluoro-4-methyl-6-(methylamino)-1,3,2-dioxoborinane (**3a**) (9 g, 55 mmol), sodium acetate (22,7 g, 276 mmol), and ethanol/water (36 mL/36 mL) was refluxed with stirring for 6 h. The solvents were evaporated, and the residue was washed several times with CH_2_Cl_2_ (25 mL). Dichloromethane was evaporated, and the residue was purified by vacuum distillation. Light yellow liquid, yield 4.56 g (72%), b.p. 95–99 °C/2–3 mbar (Ref. [46] 96–98 °C/0.13 mbar).

*N-Methyl-3-oxo-3-phenylpropanamide* (**4d**). This compound was prepared according to the procedure in Ref. [33]. The mixture of 2,2-difluoro-4-(methylamino)-6-phenyl-1,3,2-dioxoborinane (**3b**) (11.4 g, 50 mmol), sodium acetate (20.5 g, 250 mmol), and ethanol/water (50 mL/50 mL) was refluxed with stirring for 13 h. The reaction mixture was extracted with dichloromethane (2 × 100 mL). The organic layer was dried with anhydrous sodium sulphate, and the solvent was removed by distillation. White crystals, yield 7.52 g (85%), m.p. 101.5–103.5 °C (Ref. [47] 101–102 °C).

*4-Methylbenzenediazonium tetraphenylborate.* 4-Methylaniline (0.73 g, 6.84 mmol) was dissolved in boiling aqueous hydrochloric acid (3 mL, 1:1). The solution was cooled to −5 °C and diazotized by gradual addition of a cold solution of sodium nitrite (0.49 g, 7.11 mmol) in water (1.5 mL). The temperature during diazotization was maintained between −5 and 0 °C. The excess of nitrous acid (presence tested by starch iodide paper) was decomposed using the required amount of sulfamic acid. A solution of sodium tetraphenylborate (2.34 g, 6.84 mmol) in water (75 mL) was added at once. The precipitated diazonium salt was filtered and washed with cold ethanol (1 × 30 mL) and diethyl ether (2 × 30 mL). The diazonium salt was prepared immediately prior to use and dried *in vacuo* at room temperature in a desiccator for about 1 h (CAUTION: Dry diazonium tetraphenylborates can undergo violent decomposition when the crystalline material is ground!). The yield of diazonium salt was 2.81 g (94%), yellow powder.

#### General Procedure for the Reaction of **1a**–**h** with 4-Methylbenzenediazonium Tetraphenylborate

Method A. A stoichiometric amount of freshly prepared 4-methylbenzenediazonium tetraphenylborate (2.19 g, 5 mmol) was added to a stirred solution of β-enaminoamide **1a**–**h** (5 mmol) in dry dichloromethane (35 mL) at 10 °C. The reaction mixture was stirred at room temperature for 3–6 days. Then it was filtered, and the solvent was removed in vacuo. The crude residue was separated by column chromatography.

Method B. The procedure was the same as for Method A, except remelted sodium acetate (1.23 g, 15 mmol) was finely ground and added to a stirred solution of β-enaminoamide **1a**–**h** and 4-methylbenzenediazonium tetraphenylborate. The reaction mixture was stirred at room temperature for 3–6 days. Then it was filtered, and the solvent was removed in vacuo. The crude residue was separated by column chromatography.

Method C. To a cold (5 °C) solution of β-enaminoamides **1a**–**h** (5 mmol) in dry dichloromethane (15 mL) and dry toluene (25 mL) was added freshly prepared 4-methylbenzenediazonium tetraphenylborate (2.19 g, 5 mmol). The reaction mixture was stirred 30 min at room temperature and then refluxed for 3–5 h. The reaction mixture was cooled to room temperature and the solvents evaporated. The crude residue was separated by column chromatography.

*6-(1-Aminoethylidene)-2-(4-methylphenyl)-3,3-diphenyl-4H-1,2,4,3λ^4^-triazaborine-5-one* (**5a**). After column chromatography (silica gel/CH_2_Cl_2_) and boiling in ethanol (5 mL), the title compound was obtained as yellow crystals. Yield 1.16 g (61%, Method A, 72 h), 1.10 g (58%, Method B, 72 h) and 1.16 g (61%, Method C, reflux 2.5 h), m.p. 237–240 °C. ^1^H-NMR (DMSO-*d*_6_, 400 MHz): δ 12.12 (s, 1H), 10.63 (s, 1H), 7.32–7.34 (m, 4H), 7.23–7.25 (m, 2H), 7.17–7.20 (m, 4H), 7.09–7.12 (m, 2H), 6.94–6.96 (m, 2H), 6.80 (s, 1H), 2.54 (s, 3H), 2.20 (s, 3H); ^13^C-NMR (DMSO-*d*_6_, 100 MHz): δ 176.1, 160.7, 150.7 (br s), 146.0, 135.4, 133.4, 128.4, 127.0, 125.4, 123.3, 122.2, 20.5, 18.5. For elemental analysis, compound **5a** was recrystallized from toluene. Anal. Calc. for C_23_H_23_BN_4_O: C, 72.27; H, 6.06; N, 14.66. Found: C, 71.99; H, 6.13; N, 14.39%.

*6-Amino-4-methyl-5-[4-(methylphenyl)diazenyl)]-2,2-diphenyl-3H-1,3,2λ^4^-oxazaborine* (**6a**). After column chromatography (silica gel/CH_2_Cl_2_) and recrystallization from the ethanol/ethyl acetate mixture, compound **6a** was obtained as yellow crystals. Yield 0.08 g (4%, Method A, 72 h), 0.14 g (7%, Method B, 72 h), m.p. 215–218 °C. ^1^H-NMR (CDCl_3_, 500 MHz): δ 11.32 (s, 1H), 7.40–7.44 (m, 6H), 7.28–7.31 (m, 4H), 7.22–7.25 (m, 2H), 7.16–7.20 (m, 2H), 6.84 (s, 1H), 6.15 (br d, *J* = 3 Hz, 1H), 2.54 (s, 3H), 2.36 (s, 3H); ^13^C-NMR (CDCl_3_, 125 MHz): δ 170.3, 162.1, 150.4, 149.1 (br s), 137.8, 132.0, 129.7, 127.6, 126.7, 120.7, 114.6, 21.5, 21.3. Anal. Calc. for C_23_H_23_BN_4_O: C, 72.27; H, 6.06; N, 14.66. Found: C, 72.10; H, 6.21; N, 14.72%.

*6-(Amino(phenyl)methylidene)-2-(4-methylphenyl)-3,3-diphenyl-4H-1,2,4,3λ^4^-triazaborine-5-one* (**5b**). After column chromatography (silica gel/CH_2_Cl_2_) and recrystallization from the ethanol/toluene mixture, the title compound was obtained as yellow crystals. Yield 1.11 g (50%, Method A, 72 h), 0.82 g (37%, Method B, 72 h) and 1.22 g (55%, Method C, reflux 2.5 h), m.p. 133–137 °C. ^1^H-NMR (CDCl_3_, 500 MHz,): δ 12.00 (br s, 1H), 8.26 (br s, 1H), 7.42–7.44 (m, 2H), 7.32–7.36 (m, 5H), 7.21–7.27 (m, 6H), 7.20–7.18 (m, 2H), 6.99–7.01 (m, 2H), 6.74–6.76 (m, 2H), 5.18 (br s, 1H), 2.13 (s, 3H); ^13^C-NMR (CDCl_3_, 125 MHz): δ 173.3, 161.9, 149.4 (br s), 145.7, 137.0, 133.6, 133.4, 132.1, 129.8, 128.6, 128.3, 127.4, 126.1, 122.8, 122.7, 21.0. Anal. Calc. for C_28_H_25_BN_4_O: C, 75.69, H, 5.67, N, 12.61. Found: C, 75.78, H, 5.86, N, 12.60%.

*5-[(4-Methylphenyl)hydrazono]-2,2,6-triphenyl-1H,3H-1,3,2λ^4^-diazaborine-4-one* (**7b**). After column chromatography (silica gel/CH_2_Cl_2_) and boiling in cyclohexane (4 mL), the title compound was obtained as yellow crystals. Yield 0.22 g (10%, Method A, 72 h), 0.46 g (21%, Method B, 72 h), m.p. 255–260 °C. ^1^H-NMR (CDCl_3_, 500 MHz): δ 15.35 (br s, 1H), 8.46 (br s, 1H), 7.53–7.57 (m, 3H), 7.45–7.48 (m, 2H), 7.39–7.41 (m, 4H), 7.28–7.31 (m, 4H), 7.21–7.23 (m, 2H), 7.07–7.09 (m, 2H), 6.96–6.97 (m, 2H), 5.73 (br s, 1H), 2.29 (s, 3H); ^13^C-NMR (CDCl_3_, 125 MHz): δ 170.5, 164.7, 150.0 (br), 139.3, 135.7, 134.2, 132.4, 132.0, 130.2, 129.4, 128.5, 127.8, 126.5, 122.1, 116.1, 21.1. Anal. Calc. for C_28_H_25_BN_4_O: C, 75.69, H, 5.67, N, 12.61. Found: C, 75.89, H, 5.78, N, 14.46%.

*6-[1-(Methylamino)ethylidene]-2-(4-methylphenyl)-3,3-diphenyl-4H-1,2,4,3λ^4^-triazaborine-5-one* (**5c**). After column chromatography (silica gel/CH_2_Cl_2_) and recrystallization from toluene, the title compound was obtained as yellow crystals. Yield 0.85 g (43%, Method A, 72 h), 0.95 g (48%, Method B, 72 h) and 1.38 g (69%, Method C, reflux 2.5 h), m.p. 224–227 °C. ^1^H-NMR (CDCl_3_, 400 MHz): δ 13.40 (s, 1H), 7.38–7.40 (m, 4H), 7.19–7.23 (m, 6H), 7.12–7.17 (m, 2H), 6.89–6.91 (m, 2H), 5,51 (s, 1H), 3.17 (d, *J* = 5.3 Hz, 3H), 2.50 (s, 3H), 2.21 (s, 3H); ^13^C-NMR (CDCl_3_, 100 MHz): δ 174.9, 162.2, 149.8 (br s), 146.2, 136.4, 133.7, 128.6, 127.2, 126.0, 123.4, 122.8, 31.7, 21.0, 13.8. Anal. Calc. for C_24_H_25_BN_4_O: C, 72.74, H, 6.36, N, 14.14. Found: C, 73.01, H, 6.45, N, 13.95%.

*1,6-Dimethyl-5-[(4-methylphenyl)hydrazono]-2,2-diphenyl-3H-1,3,2λ^4^-diazaborine-4-one* (**7c**). After column chromatography (silica gel/CH_2_Cl_2_) and boiling in ethanol (3 mL), the title compound was obtained as yellow crystals. Yield 0.15 g (8%, Method B, 72 h), m.p. 213–216 °C. ^1^H-NMR (CDCl_3_, 400 MHz): δ 15.24 (s, 1H), 7.39–7.42 (m, 4H), 7.27–7.31 (m, 4H), 7.21–7.25 (m, 4H), 7.14–7.16 (m, 2H), 5.61 (s, 1H), 3.10 (s, 3H), 2.54 (s, 3H), 2.32 (s, 3H); ^13^C-NMR (CDCl_3_, 125 MHz): δ 171.7, 163.2, 148.5 (br s), 139.7, 134.9, 133.3, 130.2, 127.7, 126.5, 123.8, 115.7, 39.7, 21.1, 15.5. Anal. Calc. for C_24_H_25_BN_4_O: C, 72.74, H, 6.36, N, 14.14. Found: C, 73.04, H, 6.62, N, 13.85%.

*6-[(Methylamino)(phenyl)methyliden]-2-(4-methylphenyl)-3,3-diphenyl-4H-1,2,4,3λ^4^-triazaborine-5-one* (**5d**). After column chromatography (silica gel/CH_2_Cl_2_/EtOAc (10:1)) and boiling in ethanol, the title compound was obtained as yellow crystals. Yield 1.5 g (65.5%, Method A, 104 h), 1.8 g (79%, Method B, 72 h), and 1.57 g (68.5%, Method C, reflux 5h), m.p. 184–187 °C. ^1^H-NMR (CDCl_3_, 500 MHz): δ 13.22 (br q, *J* = 5 Hz, 1H), 7.39–7.45 (m, 7H), 7.25–7.26 (m, 2H), 7.18–7.21 (m, 4H), 7.11–7.13 (m, 2H), 6.89–6.92 (m, 2H), 6.65–6.67 (m, 2H), 5.7 (s, 1H), 2.88 (d, *J* = 5 Hz, 3H), 2.05 (s, 3H); ^13^C-NMR (CDCl_3_, 125 MHz): δ 174.3, 162.0, 149.6 (br s), 145.7, 136.1, 133.5, 130.4, 130.0, 128.5, 128.3, 128.2, 127.2, 125.8, 123.6, 122.2, 33.1, 20.7. For elemental analysis, compound **5d** was recrystallized from the cyclohexane/toluene mixture. Anal. Calc. for C_24_H_25_BN_4_O: C, 72.74, H, 6.36, N, 14.14. Found: C, 73.01, H, 6.45, N, 13.95%.

*1-Methyl-5-[(4-methylphenyl)hydrazono]-2,2,6-triphenyl-3H-1,3,2λ^4^-diazaborine-4-one* (**7d**). After column chromatography (silica gel/CH_2_Cl_2_/EtOAc (10:1)) and boiling in ethanol for 5 min, the title compound was obtained as yellow crystals. Yield 0.04 g (1.7%, Method A, 104 h), 0.24 g (10.5%, Method B, 72 h), and 0.22 g (9.6%, Method C, reflux 5 h), m.p. 207.5–211 °C. ^1^H- NMR (CDCl_3_, 500 MHz): δ 15.07 (s, 1H), 7.49–7.51 (m, 7H), 7.32–7.35 (m, 4H), 7.26–7.27 (m, 4H), 6.98–7.00 (m, 2H), 6.75–6.76 (m, 2H), 5.68 (s, 1H), 2.89 (s, 3H), 2.24 (s, 3H); ^13^C-NMR (CDCl_3_, 125 MHz): δ 172.6, 163.7, 148.6 (br s), 139.5, 134.9, 133.3, 133.1, 130.0, 129.6, 128.5, 127.7, 127.5, 126.6, 124.8, 115.6, 41.1, 21.0. For elemental analysis, compound **7d** was recrystallized from toluene. Anal. Calc. for C_29_H_27_BN_4_O: C, 75.99, H, 5.94, N, 12.22. Found: C, 76.20, H, 6.01, N, 12.15%.

*4-Methyl-6-(methylamino)-5-[(4-methylphenyl)diazenyl]-2,2-diphenyl-3H-1,3,2λ^4^-oxazaborine* (**6e**). After column chromatography (silica gel/CH_2_Cl_2_ or *n*-hexane/EtOAc) and boiling in ethanol (4 mL) for 5 min, the title compound was obtained as yellow crystals. Yield 0.54 g (29%, Method A, 72 h), 1.01 g (51%, Method B, 72 h), and 0.63 g (32%, Method C, reflux 5 h), m.p. 173–176 °C. ^1^H-NMR (CDCl_3_, 400 MHz): δ 11.95 (br s, 1H), 7.41–7.44 (m, 6H), 7.26–7.29 (m, 4H), 7.20–7.23 (m, 2H), 7.16–7.19 (m, 2H), 6.79 (br s, 1H), 3.11 (d, *J* = 4.4 Hz, 3H), 2.54 (s, 3H), 2.35 (s, 3H); ^13^C-NMR (CDCl_3_, 100 MHz): δ 169.4, 161.6, 150.5, 149.8 (br s), 137.2, 131.7, 129.7, 127.4, 126.4, 120.5, 115.3, 26.3, 21.2 (2 × CH_3_). For elemental analysis, compound **6e** was recrystallized from the cyclohexane/toluene mixture. Anal. Calc. for C_24_H_25_BN_4_O: C, 72.74, H, 6.36, N, 14.14. Found: C, 73.13, H, 6.39, N, 14.45%. HR-MS (MALDI), calc. for C_24_H_26_BN_4_O: [M + H]^+^ 397.2194, found: 397.2201 [M + H]^+^; calc. for C_24_H_26_BN_4_ONa: [M + Na]^+^ 419.2014, found: 419.2022 [M + Na]^+^; calc. for C_24_H_26_BN_4_OK: [M + K]^+^ 435.1723, found 435.1753 [M + K]^+^.

*1,4-Dimethyl-2,2-diphenyl-5-[(4-methylphenyl)hydrazono]-1H-1,3,2λ^4^-diazaborine-4-one* (**7e**). After column chromatography (silica gel/CH_2_Cl_2_) and boiling for a few minutes in ethanol (4 mL), the title compound was obtained as yellow crystals. Yield 0.5 g (27%, Method A, 72 h) and 0.31 g (16%, Method C, reflux 5 h), m.p. 148.5–151.5 °C. ^1^H-NMR (CDCl_3_, 500 MHz): δ 15.6 (s, 1H), 8.10 (br s, 1H), 7.40–7.41 (m, 4H), 7.30–7.33 (m, 4H), 7.22–7.27 (m, 4H), 7.16–7.18 (m, 2H), 2.68 (s, 3H), 2.48 (s, 3H), 2.35 (s, 3H); ^13^C-NMR (CDCl_3_, 125 MHz): δ 170.4, 163.7, 147.7 (br s), 139.6, 135.1, 133.4, 130.2, 127.7, 126.7, 122.9, 115.8, 30.7, 21.2, 21.1. For elemental analysis, compound **7e** was recrystallized from the cyclohexane/toluene mixture. Anal. Calc. for C_24_H_25_BN_4_O: C, 72.74, H, 6.36, N, 14.14. Found: C, 72.49, H, 6.37, N, 14.30%.

*5,N-Dimethyl-2-(4-methylphenyl)-3,3-diphenyl-4H-1,2,4,3λ^4^-triazaborine-6-carboxamide* (**8e**). After column chromatography (silica gel/CH_2_Cl_2_ → EtOAc), the title compound was obtained as orange crystals. Yield 0.51 g (26%, Method C, reflux 5 h), m.p. 184–187 °C. ^1^H-NMR (CDCl_3_, 500 MHz): δ 7.63 (br s, 1H), 7.31–7.32 (m, 4H), 7.16–7.22 (m, 8H), 7.02 (q, *J* = 5 Hz, 1H), 6.89–6.91 (m, 2H), 2.83 (s, *J* = 5 Hz, 3H), 2.53 (s, 3H), 2.21 (s, 3H); ^13^C-NMR (CDCl_3_, 125 MHz): δ 164.9, 158.8, 147.1 (br s), 145.3, 136.5, 133.6, 128.7, 127.5, 126.7, 123.0, 25.9, 23.8, 21.1. For elemental analysis, compound **8e** was recrystallized from cyclohexane. Anal. Calc. for C_24_H_25_BN_4_O: C, 72.74, H, 6.36, N, 14.14. Found: C, 72.73, H, 6.37, N, 14.39%.

*6-(Methylamino)-5-[(4-methylphenyl)diazenyl]-2,2,4-triphenyl-3H-1,3,2λ^4^-oxazaborine* (**6f**). After column chromatography (silica gel/CH_2_Cl_2_) and recrystallization from the of ethanol/toluene mixture, the title compound was obtained as yellow crystals. Yield 0.18 g (8%, Method A, 96 h), 0.98 g (43%, Method B, 72 h), and 0.48 g (21%, Method C, reflux 2.5 h), m.p. 228–230 °C. ^1^H-NMR (CDCl_3_, 500 MHz): δ 12.09 (br s, 1H), 7.61–7.63 (m, 2H), 7.49–7.52 (m, 5H), 7.44–7.47 (m, 2H), 7.28–7.31 (m, 4H), 7.21–7.23 (m, 4H), 7.08–7.10 (m, 2H), 6.84 (s, 1H), 3.20 (d, *J* = 4.9 Hz, 3H), 2.31 (s, 3H); ^13^C-NMR (CDCl_3_, 125 MHz): δ 168.8, 163.1, 150.3, 149.7 (br s), 137.4, 136.2, 131.8, 131.0, 129.8, 129.7, 128.1, 127.5, 126.5, 120.8, 115.2, 26.7, 21.3. Anal. Calc. for C_29_H_27_BN_4_O: C, 75.99, H, 5.94, N, 12.22. Found: C, 76.28, H, 6.08, N, 12.09%.

*3-Methyl-5-[(4-methylphenyl)hydrazono]-2,2,6-triphenyl-1H-1,3,2λ^4^-diazaborine-4-one* (**7f**). After column chromatography (silica gel/CH_2_Cl_2_) and recrystallization from the ethanol/toluene mixture, the title compound was obtained as yellow crystals. Yield 1.13 g (51%, Method A, 96 h), 0.2 g (9%, Method B, 72 h), and 0.3 g (13%, Method C, reflux 2.5 h), m.p. 238–240 °C. ^1^H NMR (CDCl_3_, 500 MHz): δ 15.75 (s, 1H), 8.12 (s, 1H), 7.53–7.54 (m, 3H), 7.45–7.49 (m, 6H), 7.31–7.35 (m, 4H), 7.24–7.27 (m, 2H), 7.08–7.09 (m, 2H), 6.99–7.01 (m, 2H), 2.75 (s, 3H), 2.29 (s, 3H); ^13^C-NMR (CDCl_3_, 125 MHz): δ 169.5, 164.4, 148.1 (br), 139.5, 135.3, 134.1, 133.4, 131.9, 130.2, 129.4, 128.5, 127.8, 126.6, 122.2, 115.9, 30.9, 21.1. Anal. Calc. for C_29_H_27_BN_4_O: C, 75.99, H, 5.94, N, 12.22. Found: C, 76.16, H, 6.11, N, 12.49%.

*4-Methyl-6-[1-(methylamino)ethyliden]-2-(4-methylphenyl)-3,3-diphenyl-1,2,4,3λ^4^-triazaborine-5-one* (**5g**). After column chromatography (Method A: silica gel/n-hexan → CH_2_Cl_2_/n-hexan (1:4) → CH_2_Cl_2_ → EtOAc, Method C: silica gel/n-hexane/EtOAc (15:1) → EtOAc), the title compound was obtained as yellow crystals. Yield 0.53 g (26%, Method A, 72 h), 0.56 g (30%, Method B, 72 h), and 0.99 g (48%, Method C, reflux 4 h), m.p. 193–196 °C. ^1^H-NMR (CDCl_3_, 500 MHz): δ 13.88 (br q, *J* = 4.5 Hz, 1H), 7.45–7.47 (m, 4H), 7.20–7.22 (m, 4H), 7.13–7.15 (m, 2H), 7.08–7.1 (m, 2H), 6.82–6.83 (m, 2H), 3.08 (d, *J* = 5.4 Hz, 3H), 2.57 (s, 3H), 2.43 (s, 3H), 2.16 (s, 3H); ^13^C-NMR (CDCl_3_, 125 MHz): δ 174.6, 161.1, 147.5 (br), 146.3, 136.0, 134.0, 128.3, 127.1, 125.8, 123.0, 122.8, 31.5, 30.5, 20.9, 13.6. For elemental analysis, compound **5g** was recrystallized from the cyclohexane/toluene mixture. Anal. Calc. for C_25_H_27_BN_4_O: C, 73.18, H, 6.63, N, 13.65. Found: C, 73.39, H, 6.73, N, 13.56%.

*3,4-Dimethyl-6-(methylamino)-5-[(4-methylphenyl)diazenyl]-2,2-diphenyl-1,3,2λ^4^-oxazaborine* (**6g**). After column chromatography (see **5g**) and recrystallization from the ethanol/toluene mixture, the title compound was obtained as yellow crystals. Yield 0.65 g (32%, Method A, 72 h), 0.32 g (17%, Method B, 72 h) and 0.11 g (5.4%, Method C, reflux 4 h), m.p. 215.5–221 °C. ^1^H-NMR (CDCl_3_, 500 MHz): δ 12.22 (br s, 1H), 7.41–7.43 (m, 2H), 7.38–7.39 (m, 4H), 7.27–7.30 (m, 4H), 7.22–7.25 (m, 2H), 7.16–7.17 (m, 2H), 2.99 (s, 3H), 2.64 (s, 3H), 2.35 (s, 3H); ^13^C-NMR (CDCl_3_, 125 MHz): δ 170.1, 161.3, 150.7, 148.3 (br), 136.6, 133.1, 129.7, 127.3, 126.5, 120.3, 116.2, 37.6, 26.3, 21.3, 15.2. HR-MS (MALDI), calc. for C_25_H_28_BN_4_O: [M + H]^+^ 411.2351, found: 411.2358 [M + H]^+^; calc. for C_25_H_27_BN_4_ONa: [M + Na]^+^ 433.2170, found: 433.2178 [M + Na]^+^; calc. for C_19_H_22_BN_4_O: [M + H − Ph]^+^ 333.1881, found: 333.1886 [M + H − Ph]^+^.

*1,3,6-Trimethyl-5-[(4-methylphenyl)hydrazono]-2,2-diphenyl-1,3,2λ^4^-diazaborine-4-one* (**7g**). After column chromatography (see **5g**) and recrystallization from the ethanol/toluene mixture, the title compound was obtained as yellow crystals. Yield 0.13 g (6.3%, Method C, reflux 4 h), m.p. 263.5–266 °C. ^1^H-NMR (CDCl_3_, 500 MHz): δ 15.59 (s, 1H), 7.47–7.48 (m, 4H), 7.31–7.34 (m, 4H), 7.24–7.28 (m, 4H), 7.17–7.19 (m, 2H), 3.01 (s, 3H), 2.56 (s, 3H), 2.53 (s, 3H), 2.35 (s, 3H); ^13^C-NMR (CDCl_3_, 125 MHz): δ 171.2, 163.0, 146.3 (br), 140.0, 134.6, 133.9, 130.2, 127.7, 126.6, 123.8, 115.6, 39.7, 30.5, 21.1, 15.4. Anal. Calc. for C_25_H_27_BN_4_O: C, 73.18, H, 6.63, N, 13.65. Found: C, 73.58, H, 6.64, N, 13.43%.

*N**,4,5-Trimethyl-2-(4-methylphenyl)-3,3-diphenyl-1,2,4,3λ^4^-triazaborine-6-carboxamide* (**8g**). After column chromatography (silica gel/n-hexane/EtOAc (15:1) → EtOAc) and boiling in ethanol (2 mL) for 5 min, the title compound was obtained as dark yellow crystals. Yield 0.05 g (2.4%, Method C, reflux 4 h), m.p. 186–189 °C. ^1^H-NMR (CDCl_3_, 500 MHz): δ 7.30–7.31 (m, 4H), 7.16–7.22 (m, 6H), 7.08 (br q, *J* = 4.9 Hz, 1H), 7.00–7.02 (m, 2H), 6.86–6.88 (m, 2H), 3.00 (s, 3H), 2.87 (d, *J* = 4.9 Hz, 3H), 2.65 (s, 3H); ^13^C-NMR (CDCl_3_, 100 MHz): δ 165.2, 159.5, 145.8, 145.6 (br), 135.7, 134.0, 128.9, 128.4, 127.5, 126.6, 123.4, 39.1, 26.1, 21.0, 17.8; ^11^B-NMR (CDCl_3_, 128 MHz): δ 0.98. HR-MS (MALDI), calc. for C_25_H_28_BN_4_O: [M + H]^+^ 411.2356, found: 411.2358 [M + H]^+^; calc. for C_25_H_27_BN_4_ONa: [M + Na]^+^ 433.2176, found: 433.2178 [M + Na]^+^; calc. for C_25_H_27_BN_4_OK: [M + K]^+^ 449.1915, found: 449.1918 [M + K]^+^; calc. for C_19_H_22_BN_4_O: [M + H − Ph]^+^ 333.1887, found: 333.1887 [M + H − Ph]^+^.

*4,7,8-Trimethyl-2-(4-methylphenyl)-3,3,6,6-tetraphenyl-4,6-dihydro-3H-2λ^4^,3λ^4^,6λ^4^,7λ^4^-[1,3,2]oxazaborinino[6,5-e][1,2,4,3]triazaborinine* (**9g**). After column chromatography (silica gel/n-hexane → CH_2_Cl_2_ → EtOAc) and boiling in ethanol (2 mL) for 5 min, the title compound was obtained as yellow crystals. Yield 0.11 g (4%, Method A, 72 h), m.p. 217–218 °C. ^1^H-NMR (CDCl_3_, 500 MHz): δ 7.39–7.40 (m, 4H), 7.33–7.35 (m, 4H), 7.26–7.30 (m, 6H), 7.17–7.22 (m, 6H), 7.08–7.10 (m, 2H), 6.82–6.84 (m, 2H), 3.08 (s, 3H), 2.66 (s, 3H), 2.54 (s, 3H), 2.16 (s, 3H); ^13^C-NMR (CDCl_3_, 125 MHz): δ 169.3, 155.8, 146.8 (br), 145.6 (br), 145.5, 136.2, 134.1, 132.9, 128.6, 127.6, 127.5, 126.9, 126.5, 122.9, 120.0, 38.6, 31.9, 21.0, 15.1. HR-MS (MALDI), calc. for C_37_H_37_B_2_N_4_O: [M + H]^+^ 575.3148, found: 575.3160 [M + H]^+^; calc. for C_37_H_36_B_2_N_4_ONa: [M + Na]^+^ 597.2967, found: 597.2979 [M + Na]^+^; calc. for C_31_H_31_B_2_N_4_O: [M + H − Ph]^+^ 497.2679, found: 497.2692 [M + H − Ph]^+^.

*4-Methyl-6-[(methylamino)(phenyl)methylidene]-2-(4-methylphenyl)-3,3-diphenyl-1,2,4,3λ^4^-triazaborine-5-one* (**5h**). After column chromatography (silica gel/CH_2_Cl_2_) and boiling in ethanol (5 mL) for 5 min, the title compound was obtained as yellow crystals. Yield 0.35 g (14.8%, Method C, reflux 3 h), m.p. 206–211 °C. ^1^H-NMR (CDCl_3_, 500 MHz): δ 13.80 (br q, *J* = 4.6 Hz, 1H), 7.45–7.49 (m, 7H), 7.29–7.30 (m, 2H), 7.20–7.23 (m, 4H), 7.13–7.15 (m, 2H), 6.80–6.82 (m, 2H), 6.61–6.63 (m, 2H), 2.94 (d, *J* = 5.4 Hz, 3H), 2.60 (s, 3H), 2.03 (s, 3H); ^13^C-NMR (CDCl_3_, 125 MHz): δ 174.4, 161.1, 147.4 (br), 145.7, 135.8, 133.9, 130.4, 130.2, 128.4, 128.3, 128.1, 127.2, 125.9, 123.3, 122.4, 33.1, 30.5, 20.8. For elemental analysis, compound **5h** was recrystallized from ethyl acetate. Anal. Calc. for C_30_H_29_BN_4_O: C, 76.28, H, 6.19, N, 11.86. Found: C, 76.31, H, 6.27, N, 11.68%.

*3-Methyl-6-(methylamino)-5-[(4-methylphenyl)diazenyl]-2,2,4-triphenyl-1,3,2λ^4^-oxazaborine* (**6h**). After flash chromatography (n-hexane → chloroform) and washing with cyclohexane, the title compound was obtained as yellow crystals. Yield 0.53 g (22%, Method A, 96 h), m.p. 187–189.5 °C. ^1^H-NMR (CDCl_3_, 500 MHz): δ 11.94 (br s, 1H), 7.42–7.50 (m, 7H), 7.30–7.33 (m, 6H), 7.24–7.27 (m, 2H), 6.94–7.00 (m, 4H), 3.05 (d, *J* = 4.5 Hz, 3H), 2.76 (s, 3H), 2.25 (s, 3H); ^13^C-NMR (CDCl_3_, 125 MHz): δ 171.4, 162.1, 150.4, 148.1 (br), 136.7, 134.7, 133.1, 129.5, 128.8, 128.2, 128.1, 127.4, 126.6, 120.3, 117.0, 39.4, 26.5, 21.2. HR-MS (MALDI), calc. for C_25_H_28_BN_4_O: [M + H]^+^ 473.25072, found: 473.25162 [M + H]^+^; calc. for C_25_H_27_BN_4_ONa: [M + Na]^+^ 495.2327, found: 495.2335 [M + Na]^+^; calc. for C_25_H_27_BN_4_OK: [M + K]^+^ 511.2066, found: 511.2075 [M + K]^+^.

*1,3-Dimethyl-5-[(4-methylphenyl)hydrazono]-2,2,6-triphenyl-1,3,2λ^4^-diazaborine-4-one* (**7h**). After column chromatography (silica gel/CH_2_Cl_2_) and boiling in ethanol (5 mL) for 5 min, the title compound was obtained as yellow crystals. Yield 0.37 g (16%, Method C, reflux 3 h) with m.p. 160.5–163.5 °C. After flash chromatography (n-hexane → chloroform) and boiling in ethanol, the title compound was obtained as yellow crystals. Yield 0.07 g (3%, Method A, 96 h). ^1^H-NMR (CDCl_3_, 500 MHz): δ 15.43 (s, 1H), 7.57–7.59 (m, 4H), 7.49–7.50 (m, 3H), 7.35–7.38 (m, 4H), 7.27–7.30 (m, 2H), 7.22–7.24 (m, 2H), 7.00–7.01 (m, 2H), 6.76–6.77 (m, 2H), 2.78 (s, 3H), 2.61 (s, 3H), 2.25 (s, 3H); ^13^C-NMR (CDCl_3_, 125 MHz): δ 172.1, 163.3, 146.2 (br), 139.8, 134.6, 133.9, 133.1, 130.1, 129.5, 128.4, 127.7, 127.5, 126.6, 124.8, 115.4, 41.0, 30.5, 21.0. For elemental analysis, compound **7h** was recrystallized from ethyl acetate. Anal. Calc. for C_30_H_29_BN_4_O: C, 76.28, H, 6.19, N, 11.86. Found: C, 76.56, H, 6.24, N, 11.75%.

*5,N-Dimethyl-2-(4-methylphenyl)-3,3-diphenyl-1,2,4,3λ^4^-triazaborine-6-carboxamide* (**8h**). After column chromatography (silica gel/CH_2_Cl_2_) and boiling in ethanol (5 mL) for 5 min, the title compound was obtained as orange crystals. Yield 0.51 g (21%, Method C, reflux 3 h), m.p. 207.5–209 °C. ^1^H-NMR (CDCl_3_, 500 MHz): δ 7.41–7.48 (m, 3H), 7.34–7.36 (m, 4H), 7.18–7.24 (m, 8H), 7.10–7.12 (m, 2H), 6.96 (br q, *J* = 5 Hz, 1H), 6.92–6.93 (m, 2H), 2.78 (s, 3H), 2.72 (d, *J* = 5 Hz, 3H), 2.25 (s, 3H); ^13^C-NMR (CDCl_3_, 125 MHz): δ 163.9, 159.0, 145.8, 145.4 (br s), 136.3, 134.5, 134.1, 129.38, 129.36, 128.7, 128.5, 127.6, 126.7, 126.5, 123.8, 40.6, 26.0, 21.1. For elemental analysis, compound **8h** was recrystallized from ethyl acetate. Anal. Calc. for C_30_H_29_BN_4_O: C, 76.28, H, 6.19, N, 11.86. Found: C, 76.24, H, 6.28, N, 11.75%.

*4,7-Dimethyl-2-(4-methylphenyl)-3,3,6,6,8-pentaphenyl-4,6-dihydro-3H-2λ^4^,3λ^4^,6λ^4^,7λ^4^-[1,3,2]oxazaborinino[6,5-e][1,2,4,3]triazaborinine* (**9h**). After column chromatography (silica gel/n-hexane → CH_2_Cl_2_ → EtOAc) and recrystallization from cyclohexane (or boiling in acetic acid), the title compound was obtained as yellow crystals. Yield 0.45 g (15%, Method A, 96 h), 0.11 g (3%, Method C, reflux 3 h), m.p. 216–217 °C. ^1^H-NMR (CDCl_3_, 500 MHz): δ 7.39–7.40 (m, 4H), 7.45–7.52 (m, 7H), 7.27–7.39 (m, 12H), 7.16–7.23 (m, 6H), 6.77–6.79 (m, 2H), 6.63–6.65 (m, 2H), 2.93 (s, 3H), 2.70 (s, 3H), 2.07 (s, 3H); ^13^C-NMR (CDCl_3_, 125 MHz): δ 170.3, 156.3, 146.9 (br), 145.3 (br), 145.2, 135.9, 134.0, 132.9, 132.4, 129.9, 128.6, 128.4, 128.1, 127.7, 127.6, 127.0, 126.5, 122.2, 120.8, 40.5, 31.8, 20.9. HR-MS (MALDI), calc. for C_42_H_39_B_2_N_4_O: [M + H]^+^ 637.3304, found: 5637.3318 [M + H]^+^; calc. for C_42_H_38_B_2_N_4_ONa: [M + Na]^+^ 659.3124, found: 659.3136 [M + Na]^+^; calc. for C_36_H_33_B_2_N_4_O: [M + H − Ph]^+^ 559.2835, found: 559.2851 [M + H − Ph]^+^.

*Rearrangement of oxazaborines* **6a**, **6e**–**h**. The appropriate oxazaborine **6a**, **6e**–**h** was heated at 200–250 °C, and the reaction was monitored by TLC with dichloromethane as the eluent. The residue was subjected to column chromatography on silica gel (CH_2_Cl_2_).

*5-[(4-Methylphenyl)hydrazono]-2,2-diphenyl-1H-1,3,2λ^4^-diazaborine-4-one* (**7a**). Oxazaborine **6a** (0.99 g, 2.59 mmol) was used. After 6 min at 250 °C, the crude mixture was dissolved in dichloromethane. An amount of 0.42 g was not dissolved in dichloromethane, and according to ^1^H NMR it was found that it was pure diazaborinone **7a**. After column chromatography, the title compound was obtained as yellow crystals (0.39 g). Total yield 0.81 g (82%), m.p. 263.5–266 °C. ^1^H-NMR (CDCl_3_, 500 MHz): δ 15.29 (br s, 1H), 8.39 (br s, 1H), 7.349–7.35 (m, 4H), 7.27–7.29 (m, 4H), 7.21–7.24 (m, 4H), 7.17–7.20 (m, 2H), 5.73 (br s, 1H), 2.53 (s, 3H), 2.35 (s, 3H); ^13^C-NMR (CDCl_3_, 125 MHz): δ 171.2, 164.0, 149.7 (br), 139.4, 135.6, 132.4, 130.3, 127.8, 126.6, 122.8, 116.0, 21.3, 21.1. Anal. Calc. for C_13_H_23_BN_4_O: C, 72.27 H, 6.06, N, 14.64. Found: C, 72.47, H, 5.99, N, 14.54%.

*Diazaborinone* **7e**. Oxazaborine **6e** (0.79 g, 2 mmol) was used: 25 min at 200 °C and 25 min at 250 °C. After column chromatography, the title compound was obtained as yellow crystals. Yield 0.5 g (63%).

*Diazaborinone* **7f**. Oxazaborine **6f** (0.92 g, 2 mmol) was used: 70 min at 250 °C. After column chromatography, the title compound was obtained as yellow crystals. Yield 0.45 g (49%).

*Diazaborinone* **7g**. Oxazaborine **6g** (0.82 g, 2 mmol) was used: 30 min at 250 °C. After column chromatography, the title compound was obtained as yellow crystals. Yield 0.64 g (78%).

*Diazaborinone* **7h**. Oxazaborine **6h** (0.74 g, 0.74 mmol) was used: 90 min at 230 °C. After column chromatography, the title compound was obtained as yellow crystals. Yield 0.2 g (57%).

## 4. Conclusions

The novel boron containing heterocyclic compounds—triazaborinones **5a**–**d**, **5g**, and **5h**—were synthesized from the reaction of simple β-enaminoamides and 4-methylbenzenediazonium tetraphenylborate. Other products that were formed during the reaction were oxazaborines **6a**, **6e**, **6f**, and **6g**; diazaborinones **7b**–**h**; and triazaborines **8e**, **8g**, and **8h**. Heating (250 °C, 6–70 min) of oxazaborines **6** without solvent resulted in their transformation into diazaborinones **7a**–**h** in 49–82% yields. Compound **7a** was not formed by the reaction of enaminoamide **1a** and diazonium tetraphenylborate, but it was formed by ring transformation of oxazaborine **6a**. Although *N*-phenylenaminoamides **1** (R^3^ = Ph) previously used produced diazaborinones only when there was an –NH_2_ group at position 3 of *N*-phenylenaminoamide, the enaminoamides **1** (R^3^ = H or Me) used in this work gave diazaborinones even when there was –NHMe at position 3 of the enaminoamide.

Structural, spectral, and crystallographic properties of the prepared structures were studied. Triazaborinones **5c** and **5d** and oxazaborine **6a** form dimers in the solid state. There are hydrophobic cavities in the structure of triazaborinone **5d** in which toluene as a solvent is enclosed. The measured quantum yields of the fluorescence in the solid-state of selected compounds are not very high; for diazaborinone **7a**, 14.62% is the best.

## Data Availability

The data presented are available in the manuscript Appendix A. The data files in cif format for the crystal structures in this paper can be requested from the Cambridge Structural Database. CCDC 2127610–2127615 contains the supplementary crystallographic data for this paper. These data can be obtained free of charge from The Cambridge Crystallographic Data Centre via www.ccdc.cam.ac.uk/structures (accessed on 3 December 2021).

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
