# Peer review of "Synthesis and Characterization of New Boron Compounds Using Reaction of Diazonium Tetraphenylborate with Enaminoamides"

_molecules, 2022, doi:10.3390/molecules27020367_

Round 1

Reviewer 1 Report

line 78. Write ammonia instead of ammonium

lines 59-68. The fourth paragraph of the Introduction should be rewritten for better style

Justify the choice of sodium acetate as a base (method B). Are there any differences in the ratio of products when using organic bases (for example, triethylamine or DBU)

Table 1. What do the dashes mean for method B for 1h, none of the products 5-8h were obtained?

lines 148-151 Indicate the reaction conditions in which the products were obtained 9g (4%) and 9h (3%)

line 382 There is a large difference in the chemical shift of DMSO-d6 (2.55) for calibration of spectra with literature data (2.5). For the calibration of spectra, I recommend using the published data of J. Org. Chem. 1997, 62, 7512-7515 or Org. Process Res. Dev. 2016, 20, 661-667 (https://doi.org/10.1021/acs.oprd.5b00417). (It looks strange that in the previous work of M. Svobodova et al. / Tetrahedron 68 (2012) 2052 -2060 "The 13C NMR spectra were calibrated on the central signal of the solvent multiplet (d 76.9), but in this article" CDCl3 was used as an internal standard for 13C NMR (the middle signal, δ 77.23, lines 379-380)).

General procedure for the reaction of 1a–h with 4-methylbenzenediazonium tetraphenylborate Clarify for what purpose products 5-8 after chromatography were boiled in ethanol

Figure S17. What type of 13C spectrum is it?

Reviewer 2 Report

This paper submitted by Svobodava and co-authors described methods for the synthesis of some new boron compounds and used NMR, IR, crystal structures, and UV-Vis to characterize those molecules. They also proposed a mechanism of the reactions.

Overall, this paper is well-written and the synthetic methods are nicely described. The isolation yield is acceptable and the mechanism they proposed (although it is lack of evidence) is reasonable. So I support for publication in molecules with minor revision. I believe some researchers in synthetic chemistry community will find this paper helpful.

Here are some of my concerns and questions.

1) In the abstract, please describe methods A B, and C in detail.

2) In scheme 1, please list all of the R1 function groups.

3) Figure 4, please redraw the structures.

4) this might be only me but I prefer "room temperature" over "laboratory Temperature"

5)in table 2, there must be a better way to incorporate the coupling constant in the table, maybe add a column?

Reviewer 3 Report

Interesting and useful piece of work. I recomend publication after minor amendements:

- Include in the manuscript the structures of compounds 1a-h.

- Make structrues smaller by writting Ph or p-Tol instead of drawing every single phenyl ring.

- Is all the data analysis relevant? it is quite extensive and I am not sure it all needs to be discussed with so much detail in the main paper. The discussion of data could be more brief

- Add yields and missing reaction conditions to Scheme S1 in SI

Reviewer 4 Report

Boron-containing compounds are important due to their unique biological activities and versatile transformations of the C-B bonds. Svobodova and coworkers reported the syntheses of a family of oxazaborines, diazaborinones, triazaborines, and triazaborinones from diazonium tetraphenylborates and enaminoamides, and these synthesized compounds were well characterized. These compounds should show some interesting biological activities and gain some attention from medicinal chemists. I recommend the publication of this manuscript on Molecules but major revisions should be made.

  • The manuscript is hard to read. Some details, such as X-ray data, should be put in the Supporting Information.
  • In the synthetic section, significant figures should be consistent.

For example: Line 445: 3-Amino-3-phenylprop-2-enamide 444 (1b) (8 g, 49 mmol) was heated in water (33 mL) for 4 hours.  Line 449: The solution of 3-oxo-3-phenylprop-2-enamide (3.2 g, 19.6 mmol) and 33wt% methyl-449 amine in ethanol (7.4 mL, 58.8 mmol) was heated with stirring in an Ace pressure tube at 450 135–140 °C. Line 471: N-Me-471 thyl-3-oxo-3-phenylpropanamide (4d) (7 g, 40 mmol).

Round 2

Reviewer 4 Report

The manuscirpt is now publishable.